# Putting Functional Gastrointestinal Disorders within the Spectrum of Inflammatory Disorders Can Improve Classification and Diagnostics of These Disorders

**DOI:** 10.3390/biomedicines12030702

**Published:** 2024-03-21

**Authors:** Dunja Šojat, Mile Volarić, Tanja Keškić, Nikola Volarić, Venija Cerovečki, Ljiljana Trtica Majnarić

**Affiliations:** 1Department of Family Medicine, Faculty of Medicine, Josip Juraj Strossmayer University of Osijek, J. Huttlera 4, 31000 Osijek, Croatia; dunja.sojat@gmail.com (D.Š.); mvolaric@gmail.com (M.V.); 2Department of Gastroenterology and Hepatology, University Clinical Hospital Mostar, Bijeli Brijeg bb, 88000 Mostar, Bosnia and Herzegovina; 3Department Biomedicine, Technology and Food Safety, Laboratory of Chemistry and Microbiology, Institute for Animal Husbandry, Autoput Belgrade-Zagreb 16, 11080 Belgrade, Serbia; tkeskic@istocar.bg.ac.rs; 4Department of Physiology and Immunology, Faculty of Dental Medicine and Health, Josip Juraj Strossmayer University of Osijek, Crkvena ulica 21, 31000 Osijek, Croatia; nikola.volaric@fdmz.hr; 5Department of Family Medicine, School of Medicine, University of Zagreb, Šalata 3, 10000 Zagreb, Croatia; venija.cerovecki@mef.hr

**Keywords:** gastrointestinal diseases, functional gastrointestinal disorders, chronic diseases, cytokines, gastrointestinal microbiome, inflammation

## Abstract

The spectrum, intensity, and overlap of symptoms between functional gastrointestinal disorders (FGIDs) and other gastrointestinal disorders characterize patients with FGIDs, who are incredibly different in their backgrounds. An additional challenge with regard to the diagnosis of FGID and the applicability of a given treatment is the ongoing expansion of the risk factors believed to be connected to these disorders. Many cytokines and inflammatory cells have been found to cause the continuous existence of a low level of inflammation, which is thought to be a basic pathophysiological process. The idea of the gut–brain axis has been created to offer a basic framework for the complex interactions that occur between the nervous system and the intestinal functions, including the involvement of gut bacteria. In this review paper, we intend to promote the hypothesis that FGIDs should be seen through the perspective of the network of the neuroendocrine, immunological, metabolic, and microbiome pathways. This hypothesis arises from an increased understanding of chronic inflammation as a systemic disorder, that is omnipresent in chronic health conditions. A better understanding of inflammation’s role in the pathogenesis of FGIDs can be achieved by clustering markers of inflammation with data indicating symptoms, comorbidities, and psycho-social factors. Finding subclasses among related entities of FGIDs may reduce patient heterogeneity and help clarify the pathophysiology of this disease to allow for better treatment.

## 1. Introduction—Motivation for This Review

Up to today, inflammatory bowel diseases (IBDs), ulcerative colitis (UC), and Crohn’s disease (CD), which are classified within autoinflammatory diseases, have been considered the only chronic inflammatory diseases that affect the gastrointestinal tract. The increasing evidence suggests that the development of these diseases may be triggered by a disturbance in the balance between the gut commensal microflora and the mucosal immune system [1]. A discovery that colonization of the gastric mucosae by the microorganism Helicobacter pillory (HP), and the inflammation associated with this infection, underlie the most frequent disorders of the gastrointestinal tract, chronic gastritis and ulcer disease, has changed our perception of these common human diseases, for which impaired secretion of gastric acid had been considered the major etiological factor [2].

The Rome criteria were founded to enable the diagnosis of a large set of functional gastrointestinal disorders (FGIDs) to which little attention had been devoted previously [3]. The routine diagnostic examinations of these illnesses do not reveal any underlying structural defects that can explain the symptoms. The Rome classification is based on symptom clustering, taking into account the regional anatomy of the gastrointestinal tract (esophageal, gastroduodenal, bowel, biliary, and anorectal). The last update of this classification (Rome IV, 2016) recognizes 33 adult and 17 pediatric disorders. The most common adult disorders are irritable bowel syndrome (IBS) and functional dyspepsia (FD).

Although the Rome classification has led to better characterization of these disorders and improved clinical practice, it is still insufficient to enable individualized treatment, as well as accurate patient selection for clinical trials. Patients with these disorders are exceedingly diverse, as their symptoms range in scope, severity, frequency, and duration, and there is overlap between different FGIDs, as well as between these disorders and other gastrointestinal diseases. There are no suitable operational definitions or biomarkers for their accurate diagnosis [4]. Additionally challenging, concerning the relevance of the diagnosis and treatment, is the fact that the number of the risk and pathophysiology factors, that are thought to be associated with these disorders constantly expand as new evidence becomes available. For instance, in addition to increased visceral sensitivity and altered motility, as has long been believed, decreased intestinal microbiota diversity and mucosal immune system activation have also been linked to the development of these disorders [5].

In addition, epidemiological observations that altered emotions frequently co-occur with these disorders have been supported by experimental research. The concept of the gut–brain axis has been developed to provide a general framework for the intricate interactions between the nervous systems and intestinal functions, which include the gut microbiota [6,7]. A view has emerged, including a bio-psycho-social perspective, that provides a comprehensive understanding of FGIDs by trying to explain how a large array of environmental, psychological, and biological factors contribute to the expression of symptoms of these disorders [8].

In this paper, we provide an overview of the evidence that FGIDs, notably referring to FD and IBS, should be seen through the perspective of the network that integrates the neuroendocrine, immunological, and metabolic pathways, including their linkages to the gut microbiome. This view arises from an increased understanding of chronic inflammation as a systemic disorder that is omnipresent in chronic health conditions. We propose that a better understanding of the role of inflammation in the pathogenesis of FGIDs can be achieved by clustering markers of inflammation with data indicating symptoms, comorbidities, and psycho-social factors. This method, by finding subclasses within one entity or between multiple related entities of FGIDs, may reduce patient heterogeneity and help clarify the pathophysiological pathways connected with these disorders.

## 2. Autoinflammatory and Autoimmune Diseases

It has been known for a long time that inflammation plays a significant role in the pathogenesis of some diseases which often show a strong genetic influence, including musculoskeletal rheumatic diseases, IBS, psoriasis, and systemic autoimmune diseases [9,10]. Historically, they have been divided into two categories, autoinflammatory and autoimmune diseases, based on the differences in the underlying immune mechanisms. Autoinflammatory diseases are characterized by abnormal innate immune responses without high-titer autoantibodies or autoreactive T lymphocytes [11]. On the contrary, the hallmark of autoimmune diseases is a failure of the specific (adaptive) immune system to differentiate self-antigens from foreign antigens (self/nonself discrimination) [9]. However, a necessary precondition for the promotion of autoreactive T and B cells and the progression of tissue and organ damage, once the tolerance to autoantigens has been lost, is the activation and abnormal proliferation of innate immune cells, macrophages, neutrophils, and DCs, which are capable of secreting a large amount of different inflammatory mediators, such as TNF-α, interleukins (ILs), and interferons (IFNs).

The key step in this process is the establishment of the permanent imbalance between the regulatory T cells (Treg cells) and the effector T cells (Teff cells), which can develop under the long-term influence of environmental factors or immune disturbances caused by preexisted pathologies [9,12,13]. An important fact to know is that Treg cells show a high degree of plasticity under altering micro-environmental conditions [14]. This feature enables them to switch from exhibiting anti-inflammatory and tissue-safeguarding qualities to effector phenotypes and functions, hence assisting in the maintenance of the established type of immune response (either Th1 or Th2) as long as the pro-inflammatory signals disappear. While it is a useful homeostatic mechanism in the case of infection, in conditions of “sterile inflammation”, such as in obesity or the presence of chronic diseases, this characteristic of Treg cells can lead to autoimmune diseases [15].

The knowledge of inflammation-mediated diseases has been improved markedly, and it has become clear that many of these diseases consist of a mix of autoinflammatory and autoimmune components. For example, IBDs, which are primarily considered autoinflammatory diseases, often coexist with other autoinflammatory and autoimmune conditions, like ankylosing spondylitis, rheumatoid arthritis, psoriasis, primary sclerosing cholangitis, uveitis, episcleritis, celiac disease, and systemic lupus erythematosus [16]. An immunological continuum has been proposed according to which these diseases are classified depending on the degree to which the adaptive vs. innate immune system is involved in their pathogenesis [17].

A better understanding of the pathogenesis of these diseases has been enabled by deciphering the role of neutrophils in chronic inflammation associated with organ damage. In response to infection, neutrophils serve as the first line of cell-mediated defense [18]. They are short-lived cells whose role is to achieve a quick dilution of pathogens in the invaded tissue before macrophages engage in their phagocytic and tissue-repairing functions. For this purpose, they are equipped with several strong, nonspecific defense mechanisms, which, in turn, can cause substantial damage to the surrounding tissue. These mechanisms include both intracellular (phagocytosis) and extracellular mechanisms, such as releasing the various cytotoxic enzymes from their granules (degranulation) or forming neutrophil extracellular traps (NETs) [18,19]. NETosis is a cell program of the entrapping and extracellular killing of microorganisms and is associated with the release of granule content into the cytosol, histones modification, chromatin decompensation, and the formation of pores in the cell membrane. It may involve a lytic type of programmed cell death, but the expulsion of nuclear and granule contents into the extracellular space may proceed without affecting the cell membrane integrity. The neutrophil defense mechanisms also include the production of reactive oxygen species (ROS), which can kill microorganisms both intracellularly and extracellularly and can help in the formation of NETs, as well as pro-inflammatory cytokines and chemokines, whose roles are to recruit additional immune cells [18,19,20].

The same mechanisms of neutrophil-associated immune functions, although somewhat dysregulated, are implicated in the pathogenesis of chronic inflammatory diseases, including metabolic diseases [21,22,23]. In particular, the ability of neutrophils, at least some of their subpopulations such as low-density granulocytes (LDGs), to produce high levels of type I interferons (INFIa) and to form NETs, which have been found to endow them with super-strong pro-inflammatory activity and tissue pathogenic properties, are implicated in this pathogenesis. Importantly, NET formation is associated with protein, histones, and nucleic acid molecular modification by chemical processes such as oxidation, citrullination, and carbamylation, which can stimulate the production of autoantibodies and the development of autoimmune diseases. Thus, the excessive formation and/or a defective elimination of NETs is considered to lead to the development of tissue pathologies associated with inflammatory and autoimmune diseases.

## 3. The Role of Inflammation in the Pathogenesis of Other Chronic Diseases

Two findings contributed greatly to our knowledge of the role of inflammation and immunological mechanisms in the pathogenesis of chronic diseases. The first was that various stimuli, not just microorganisms, can activate the innate immune receptors, the so-called “pattern recognition receptors” (PRRs), which initiate the increased production of pro-inflammatory cytokines, such as tumor necrosis factor-α (TNF-α), IL-1β, IL-18, IL-6, and IFNs [24]. Namely, the innate immune cells, mainly including macrophages, dendritic cells (DCs), and neutrophils, contain several classes of PRRs that can be placed on the cell surface or in the intracellular space and by which they can recognize the presence of pathogens, through ligation with “pathogen-associated molecular patterns” (PAMPs), and also the endogenous “danger” signals, through ligation with “danger-associated molecular patterns” (DAMPs). Danger signals may include components of damaged cells and tissues, metabolic intermediates, ROS, and molecules indicating unfavorable conditions in the microenvironment, characterized by hypoxia or nutrient deprivation. Activation of PRRs translates the danger signals into the pro-inflammatory pathways, which, by molecules involved in signal transduction such as adaptor proteins, protein kinases, and transcription factors, finally leads to the transcription of a large array of genes that are involved in mediating inflammatory and immune responses. The inflammasome, a multi-protein complex that assembles in the cytoplasm after the activation of PRRs, plays a role in the integration and amplification of multiple danger signals through the recruitment and activation of the enzyme Caspase-1. The activated Caspase-1 splices proIL-1β/proIL-18 into the corresponding mature cytokines, whose role is to amplify the innate immune response initiated by PRRs [25].

Regardless of the type of the initial stimuli, inflammatory conditions can be established in the microenvironment which are necessary for Teff cell activation and can eventually lead to the loss of tolerance to autoantigens and the cooperation of the innate and adaptive (specific) immune responses in promoting tissue damage and remodeling [9]. Which type of immune response will dominate, either a type 1 (cell-mediated) immune response, which involves innate immune cells, like DCs, macrophages, natural killer (NK) cells, and T helper cells type 1 (Th1 cells), or a type 2 (humoral, antibody-mediated) immune response, which is characterized by the activation of Th2 and B cells, antibody production, and tissue infiltration by eosinophils and mast cells, will depend on the local cytokine milieu and an individual’s genetic makeup [11,12].

The second important finding that significantly improved our understanding of the role of inflammation in tissue damage in chronic diseases was the identification of the Th17 subtype of Teff cells [20]. The cytokines that are preferentially generated by this lymphocyte subset are cytokines of the IL-17 family, IL-21, and IL-22. These cytokines were shown to increase inflammation through the recruitment of immune cells (lymphocytes) and inflammatory effector cells, including monocytes/macrophages, neutrophils, eosinophils, mast cells, and basophils, from circulation to the site of inflammation where they then play a role in the elimination of detritus, tissue repair, and remodeling [26,27]. In addition to Th17 lymphocytes, the innate and inflammatory cells that are expanded in inflamed tissue, including neutrophils, become capable of producing the IL-17A cytokine, which is a mechanism for maintaining chronic inflammation and continuous neutrophil recruitment [28]. Neutrophils have a prominent role in the tissue damage/repair processes, not only in diseases classified as autoinflammatory and autoimmune, but also in common chronic conditions associated with obesity and metabolic disorders, like hypertension, type 2 diabetes (T2D), metabolic syndrome, and cardiovascular disease (CVD) [23,27,29].

Thus, Th17 cells appear to be crucial in maintaining the chronicity of the inflammatory response. The balance between tissue damage and tissue repair/fibrosis is thought to be regulated by oscillation in the predomination of either Treg cells or Th17 cells, which can depend on the net effect of tissue-related and systemic signals. For example, low concentrations in the tissue of the transforming growth factor-β (TGF-β), combined with the presence of cytokines IL-6 or IL-23, promote Th17 cell development and a pro-inflammatory microenvironment. Conversely, when there are higher TGF-β concentrations this overrides the pro-inflammatory stimuli by shifting the Treg/Th17 balance towards the Treg cell predomination and promoting tissue repair/fibrosis, which in turn can also be detrimental to maintaining the tissue architecture and function. This oscillating dynamic is possible due to the plasticity of both Treg and Th17 cells, and some other immune cells, in particular macrophages, which can oscillate between the pro-inflammatory M1 and reparatory M2 phenotype depending on the conditions [30,31].

## 4. Inflammation in the Vital Tissue and the Whole Body’s Reaction

Inflammation has been initially defined as an evolutionarily conserved response that protects the host from invading microorganisms and tumors while promoting tissue repair following injury [32]. Subsequently, it has been recognized that the immune system regulates a wide range of physiological processes, including neurological and gastrointestinal system function, metabolism, thermogenesis, and tissue regeneration and remodeling [32,33]. In addition, it is also involved in all types of homeostasis perturbations that may be caused by different factors, including changes in diet and environmental temperature, emotional disturbances, sleep deprivation, exposure to infections, toxins, and injuries.

Thus, inflammation can be understood as a multistep process that involves mobilizing defense mechanisms to eradicate the cause of homeostasis disruption. This reaction may range from the reparatory physiological responses that occur in the absence of tissue damage to acute, time-limited reactions to infections or noxious injury which can terminate with restitutio ad integrum or adaptive tissue remodeling but can also eventually lead to a rapidly progressive homeostasis breakdown, in the form of sepsis. Finally, there is the potential for chronic inflammation, which is usually driven by non-infectious causes, to lead to the gradual deterioration of homeostasis which can eventually result in the development of chronic diseases [32,33,34]. A view is emerging that inflammation is linked to almost all human diseases [35].

## 5. An Interplay between the Neuroendocrine, Immune, and Metabolic Pathways in Aging and Obesity as a Driver of Chronic Disease Development

Unlike acute inflammation, which is often restricted to the local tissue environment and does not involve the whole body’s (systemic) reactions, chronic inflammation always involves systemic metabolic and neuroendocrine changes that appear along with alterations in the structure and functions of a variety of tissues and organs [35]. These impacts may occasionally only include functional changes rather than structural changes [32,33]. The heterogeneity of pathology, and variations in the dynamics of the tissue damage, may depend not only on the type and magnitude of inflammatory response but also on the responsiveness/resistance of target tissues to inflammatory challenges, as well as the ability of control mechanisms to counteract the negative cost of inflammation and restore the body’s balance [33,36]. The period of time until which the adaptive control mechanisms can buffer the deleterious effects of inflammation-related mechanisms on tissues is determined by an individual’s genuine protective capacity and the degree to which the homeostatic mechanisms are undermined by past challenges [37].

Understanding how inflammation plays a role in the emergence of chronic diseases from a systemic viewpoint would aid in comprehending why these conditions usually manifest as several disorders coexisting together, and how their pathology gradually spreads over time [32,33,34]. To help clarify this view, we will mention the theory of the “immunological homunculus” (“Immunculus”). This theory proposes that screening the adult population, in particular individuals with some risk factors for common chronic diseases, on their serum content of constitutively expressed natural antibodies (naAbs) could enhance the early diagnosis of health disturbances [38]. As proposed by this theory, the human immune system can produce a huge amount of naAbs (low-affinity immunoglobulins of the “M”, “G”, and “A” classes) directed to the multitude of self-antigens which participate in maintaining homeostasis of the human body, this is considered a complex system. Namely, the high level of complexity of the human body proposes the existence of regulatory systems and their assistance in preserving molecular (antigenic) integrity and maintaining the strict regulation of intercellular and intersystem communications. The naAb repertoire is established during ontogenesis, is influenced by the maternal immune imprinting, and remains constant until adulthood as long as a healthy state is maintained. The accumulating evidence indicates that it undergoes changes in different pathologic states and that, by identifying the patterns of these changes, it may become an instrument in the early (pre-clinical) diagnosis of disturbances in the body’s functional state.

Considering the immunologic system as a self-regulatory (auto-reactive) system rather than as having a role in recognizing foreign antigens, as stated by the classical immunologic theories, might have important practical implications [39]. It may explain the close interrelations between the immunologic system and two other homeostasis-regulating systems, the glucose-dependent metabolism system and the neuroendocrine system, so that disturbances in one system may cause changes in the other two [40]. This can provide a better understanding of the “inflame-aging” theory, which states that a reduction in the capacity to cope with a variety of stressors (where an increase in antigenic load is considered a type of stressor), and a concomitant increase in systemic inflammation, is a major characteristic of the aging process [41]. Observations on the close association between metabolic syndrome and autoimmune diseases, as well as the diversification of the immunologic reaction in elderly individuals, both as suppression (manifested by low surveillance of infections and increased risk for malignant diseases) and increased susceptibility to autoimmune reactions, can be placed within this concept [15,42]. There is a belief that strategies that focus on restoring the “benign auto-reactivity” by bringing back deviations in immunologic reactions within the boundaries of the normal intensity could set a new standard in curing immunologically mediated diseases [43]. This view on the integrating role of the immune system in maintaining the body’s homeostasis is further elaborated in the following paragraphs.

Epidemiologic research indicates that the frequency and complexity of chronic diseases increase with age, potentially due to a decline in the body’s ability to maintain homeostasis, which is caused by the “wear-and-tear” of cells and tissues. The immune system’s prolonged exposure to harmful stimuli, which occurs concurrently, causes a permanent rise in systemic inflammation, hastening the pathophysiological alterations throughout the body [41]. Chronically active innate immunity, a shift from specific immunity toward non-specific immunity, and autoimmune reactions are characteristics of the aging immune system. These factors collectively lead to a decrease in the development of specific immune responses to foreign antigens [44]. The early emergence of comorbidities accelerates aging and the development of poor health-related outcomes, although older individuals can follow different aging trajectories depending on lifestyle choices and life conditions they encounter throughout their lives [45].

The “sickness phenotype”, which manifests in acute inflammation when pro-inflammatory cytokines alter neural circuits, may also accompany chronic inflammatory conditions in a modified form, exhibiting symptoms like fatigue, depression, low activity, altered sleep, muscle wasting, and social withdrawal. These observations lend support to the idea that chronic inflammation is a systemic disorder [33,35]. This phenotype is thought to be the neuroendocrine system’s homeostatic reaction to increased energy demand due to continuous immune system activity, which requires fuel allocation from energy storage units to the immune cell compartment [46]. This metabolic derangement is connected with insulin resistance (reduced efficiency of insulin in glucose utilization in insulin-sensitive tissues, such as muscle, adipose tissue, the liver, and the brain).

Insulin resistance is an adaptive homeostatic mechanism that operates at the interface of metabolic derangements and inflammation. In the long run, it is always maladaptive, having deleterious effects on health [44]. For instance, obesity is a condition associated with insulin resistance that increases the risk of developing many chronic diseases, including adult asthma, osteoarthritis, and certain types of cancer, as well as cardio-metabolic diseases like T2D, hypertension, CVD, neurodegenerative diseases, and nonalcoholic fatty liver disease [47,48]. Adipose tissue in obese individuals is a source of inflammation, as it is abundantly infiltrated by macrophages. This is an adaptive mechanism designed to counteract the excessive build-up of calories (in the form of lipids) in adipose tissue through to phagocytosis of lipids by macrophages. The majority of pro-inflammatory cytokines, however, are produced by activated macrophages. Insulin resistance is a good adaptation strategy that seems to minimize subsequent calorie storage and additional adipose tissue inflammation [47]. However, as time goes on, a vicious cycle of homeostasis breakdown occurs, involving mechanisms like tissue and cell resistance to adrenergic (sympathetic) stimulation, as well as the persistent activation of the hypothalamic-pituitary-adrenal (HPA) stress axis. This, combined with a disrupted circadian rhythm, may exacerbate a number of pathophysiology pathways and accelerate the progression of tissue and organ pathology (Figure 1) [33,46,47,48].

The importance of obesity as a source of inflammation and a driving force for the development of a wide range of chronic diseases requires a more detailed discussion. Mounting data suggest that obese people demonstrating the abdominal type of obesity have a higher level of insulin resistance than those with general obesity and are more likely to have atherosclerotic CVD [49]. In these individuals, the visceral adipose tissue is abundantly infiltrated with immune cells associated with type 1 inflammation, such as NK and NKT (share properties of NK and T cells) cells, innate lymphoid cells type 1 (ILC1), and Th1 cells. This phenotype is characterized by pro-inflammatory macrophage polarization (M1) and increased production of pro-inflammatory cytokines, including TNF-α and IL-6 (Figure 2) [33,47,50]. A variation in the degree of adipose tissue inflammation may be utilized to distinguish metabolically unhealthy from metabolically healthy obese patients [49].

Interestingly, interventions that are known to lessen the pathological potential of obesity, like physical activity or a healthy diet, have been shown to reverse type 1 to type 2 inflammation in the adipose tissue of obese individuals—this process is termed “browning of the white adipose tissue” [51]. Type 2 inflammation is mediated by ILC2 and Th2 cells and the M2 type of macrophages, and involves tissue infiltration with eosinophils and mast cells, along with the secretion of type 2 cytokines, such as IL-13, IL-4, and IL-5 (Figure 2). Many details associated with this process, which otherwise might lead to efficient interventions, are not yet clear. In particular, there is a need to clarify the role of obesity in the pathogenesis of CVD and other chronic diseases in women and men, respecting the fact that men are more prone to atherosclerotic CVD, which is associated with type 1 inflammation, while women are more prone to type 2 inflammation and Th2/Th17 cell-mediated immune reaction [30,52,53].

## 6. The Role of the Gut Microbiome and the Gut Mucosal Immune System in the Development of Chronic Disease

The detailed presentation of the interplay between the gut microbiome and the gut mucosal immune system in the development of chronic diseases is out of the scope of this review. Yet, it is necessary to mention it briefly because of the growing evidence indicating the presence of dysbiosis (changes in the composition and diversity of bacterial communities in the gut) and changes in gut mucosal immunity in different pathological conditions, when compared to healthy controls (Figure 1) [54].

The gut microbiome is a large collection of microorganisms that inhabit the intestinal lumen and is considered a functioning organ that plays essential roles in human physiology. It is the most important among the microbial ecosystems that colonize the skin and mucosal surfaces of the body in terms of determining the trade-offs between health and disease since it is the largest and most complex [55]. Its biological potential is illustrated by the fact that the number of cells in the gut microbiome exceeds the total number of cells in the human body by a factor of more than ten [56].

The process of colonization of the intestine (and other mucosal surfaces) begins at birth and lasts until 2–3 years of age when the number and the composition of microbial families stabilize. The exposure of the gut mucosal innate immune receptors to microbial components provides necessary signals for postnatal immune system maturation, including both mucosal and systemic immunity [55]. If the conditions for gut microbial colonization are not favorable, like in premature infants, when a baby is not nursed, or when antibiotics are used, this will have long-term negative consequences on health (Figure 1) [57]. Conversely, developing the gut microflora under unobtrusive conditions establishes its symbiotic or beneficial relationships with the host, which is reflected in the term “commensal microflora”. These beneficial effects are achieved through several routes: (1) by protecting the host against infections, (2) ensuring tolerance to foods and the microflora itself, (3) contributing to nutrient digestion and the extraction of some essential nutrients from food that otherwise cannot be extracted, and (4) by keeping the intestinal epithelial barrier intact [58].

Later during life, under the influence of unfavorable environmental factors such as an unhealthy diet, antibiotic use, psychological stress, and exposure to toxins, especially in genetically susceptible individuals, this beneficial equilibrium may turn detrimental, leading to the development or worsening of chronic diseases (Figure 1) [59].

The commensal microflora regulates immune functions by providing microbe-associated molecular patterns, like lipopolysaccharides (LPS) and peptidoglycans, that serve as antigens for innate immune cell activation when microbial components penetrate the mucosal epithelial barrier [60]. In addition, by providing microbial metabolites, such as short-chain fatty acids (SCFAs), branched-chain amino acids (BCFAs), tryptophan metabolites, butyrate, propionate, and acetate, the commensal microflora mediates metabolism and, thus, also the activity of the immune cells [61]. One more way is indirect, via the fine-tuning of neuroendocrine mechanisms, including the enteric and autonomic nervous systems and the CNS, which in turn can modulate the activity of immune cells via receptors for neurotransmitters and hormones on their surface (Figure 1) [62].

The mucosal surfaces of the respiratory, gastrointestinal, and urogenital tract are the areas where the body comes into contact with the external world, and where most pathogens, but also harmless antigens, such as food and airborne antigens, enter the body [58]. The mucosal compartments, in particular the intestine, have developed structurally and functionally complex immune systems that are capable of mounting immune responses against pathogens while maintaining tolerance towards non-pathogenic antigens, including those originating from the commensal microflora.

This flexibility of the gut mucosal immune system is also visible in situations of the excessive growth of gut microbes or when more aggressive strains with higher inflammatory potentials threaten to become prevalent among microbial communities and breach the mucosal epithelial barrier. In this case, the gut mucosal immune system may impose selective pressure on microbial strains by varying the types and intensity of defense mechanisms and the degree to which innate and adaptive immune responses are being engaged (Figure 2) [58,60]. Thus, flexibility has the purpose of avoiding the overwhelming inflammation of the gut mucosa and its impact on increasing systemic inflammation.

It is thought that because of the flexibility of the mucosal immune system, and the heterogeneity of the observed inter-individual immune responses, there is the added benefit of a spatial diversity of microbial species [63]. It is observed to exist along both the longitudinal (the stomach, the small intestine, the cecum, and the colon) and the transversal axis (the luminal, the crypt-associated, and inner mucus layers—associated microbiota) of the gastrointestinal tract.

To fulfill its complex function, the gut mucosal immune system is structured in a way that inductive and effector sites are spatially separated (Figure 2) [58,64]. Inductive sites consist of organized lymphoid structures, lymphoid follicles, and Peyer’s patches, which are mostly situated in the subepithelial space of the ileum. The effector sites include a variety of individual immune cells diffusely distributed throughout the lamina propria of the colonic mucosae.

As can be seen in Figure 2, the intact epithelial cell and mucus layers, innate immune cells that can change their phenotypes along with changing local conditions, and secretory immunoglobulin A (IgA) antibodies, form the basis of the mucosal immune response [49,56]. The IgA antibody response is highly flexible [65]. These antibodies are primarily synthesized in the professional inductive sites, the lymphoid follicles and Peyer’s patches, where antigen-producing B cells can come into contact with follicular dendritic cells (FDCs) and follicular Th (Thf) cells, which allows for the induction of the hypermutation of their immunoglobulin genes and the selection of high-affinity antibodies. This is a situation when IgA antibodies are produced during the specific immune response, usually in response to pathogens [65,66]. These antibodies can also be produced as a part of the innate immune response, where Th cells are not necessary for their production [44]. In this case, IgA antibodies are of low affinity and restricted diversity, and their function is to sustain commensal microflora in a homeostatic state [65].

The preferred way in which the mucosal immune system influences the development of various chronic diseases is by increasing the system’s level of inflammation, the mechanisms of which we describe in the next two paragraphs. Another way is through the effects of the gut microbiota and mucosal immune system in modulating the activity of the gut–brain axis by releasing a variety of neuroactive metabolites [67]. We provided detailed information on this in our recently published paper [68].

In a healthy condition, the immune response to intestinal microbiota is focused on the mucosal surface (Figure 2) [49,52,56]. The mechanisms that prevent the epithelial barrier from leaking microbial components include intact epithelium and mucus layers, a range of antimicrobial peptides generated by specialized epithelial cells, as well as an efficient IgA antibody response (Figure 2). DCs and IgA antibodies play a critical role in sampling the gut microbes. Sensing the gut microbiota via innate cell receptors in the lamina propria mucosae results in the low-grade activation of innate immune cells (priming). This is an adaptive mechanism that maintains microbial populations in equilibrium and holds active mechanisms for safeguarding the epithelial barrier.

In situations when pro-inflammatory signals from the gut microbiota become stronger, in the presence of gut microbiota perturbations, the level of activation of innate immune cells increases, creating conditions for the establishment of inflammation, and, thus, also creating conditions for initiating the specific immune reaction. ILCs play an important role in controlling the level of inflammation and facilitating the transition from “physiologic” to “pathologic” inflammation. These cells lack surface antigen-dependent receptors but can synthesize the respective types of cytokines by which they initiate the differentiation of naive Th cells into Th1, Th2, or Th17 subtypes (Figure 2).

## 7. Functional Gastrointestinal Disorders

FGIDs are the most prevalent diagnoses in gastroenterology, defined by the persistence of unpleasant symptoms of the gastrointestinal tract in the absence of obvious structural abnormalities [69]. More than 40% of individuals in the general population suffer from FGIDs, which affects their quality of life and increases their use of healthcare services. Recognition of FGIDs has expanded in recent years thanks to the ROME classification, which is based on the clustering of symptoms, but the pathogenesis of these disorders remains unclear [70,71]. As potential pathophysiology factors, visceral hypersensitivity, dysfunctional gut motility, post-infectious gastroenteritis, increased intestinal permeability, altered gut microbiota, irregular gut–brain connection, and chronic low-grade intestinal mucosal inflammation, have been recognized [72,73,74,75,76,77]. They are also characterized by a high prevalence of mental comorbidities and chronic pain problems, which explains a relationship between the gastrointestinal tract and the brain (the gut–brain axis) for which reason these disorders have, in recent times, been considered gut–brain interaction disorders [74,78]. FD and IBS are the most common FGIDs [70].

FD is the most prevalent FGID, with a prevalence of 20% to 40%, accounting for 3% to 5% of primary care visits. People with FD have a poor quality of life due to persistent or recurring upper abdomen pain or discomfort which may be compared with that of patients with moderate heart failure [75]. Using the Rome IV criteria, FD can be classified as postprandial distress syndrome (PDS) or epigastric pain syndrome (EPS), with possible overlapping. The first subcategory is defined as an uncomfortable early satiation and/or postprandial fullness, and the second subcategory is defined as discomfort and/or burning. FD frequently coexists with other FGIDs, especially IBS [71].

IBS is a complex disorder that causes persistent stomach discomfort and irregular bowel movements. Symptoms may overlap with those of other FGIDs, and up to one-third of individuals with FGIDs exhibit characteristics of several conditions, indicating a shared underlying cause [74]. According to the Rome IV criteria, patients are classified into four categories based on their predominant bowel habit: diarrhea-predominant (IBS-D), constipation-predominant (IBS-C), mixed diarrhea/constipation (IBS-M), and unclassified (IBS-U) [79]. IBS has a predicted prevalence of 10–15% and is one of the leading causes of primary care visits, resulting in increased healthcare expenditures [70].

## 8. An Association of Inflammation with FGIDs Including the Role of the Gut Microbiome

The gut–brain axis appears to be impacted by disruptions in the complex community of bacteria. Studies revealed that patients with FGIDs had higher levels of T lymphocytes, mast cells, eosinophils, and macrophages in their guts. Tryptase, histamine, cytokines, and prostaglandins, which are released by activated mast cells, have been linked to changes in nociceptive pathways and intestinal barrier functioning, causing the barrier to become leaky [79,80]. Abnormalities in tight junctions caused by changes in the gut microbiome can promote chronic inflammation leading to an increase in the intestinal mucosa’s bacterial and byproduct leakage, which is connected to non-specific and specific immune reactions [81]. Furthermore, there is indirect evidence that Th17, in combination with Th2 immunological responses, has an important role in the etiology of FGIDs [69,74].

Extensive research indicates that patients suffering from FGID exhibit a cytokine imbalance in the systematic circulation and in the intestinal mucosa. These findings support the notion that inflammation plays a crucial role in the pathogenesis of these disorders. Currently, studies cannot distinguish between an isolated intestinal wall inflammation and elevated bloodstream inflammatory markers resulting from systemic diseases and chronic disorders as the etiology of FGIDs [82].

One more way in which the gut microbiota regulates immune functions is via neuroendocrine mechanisms, including the enteric nervous system (ENS), autonomic nervous system (ANS), and CNS, as well as soluble factors, like neurotransmitters, neuropeptide hormones, and interleukins [50]. The intestinal enterochromaffin cells (ECCs) secrete 5-hydroxytryptamine (5-HT) as a result of the bacteria in the gut microbiome producing SCFAs that activate the enzyme tryptophan hydroxylase 1 (TPH1). Intestinal ECCs emit 5-HT, which binds to receptors on ENS neurons to alter intestinal motility [83].

In order to control some essential GI functions, a variety of unique ion channels or their subtypes can target different GI cells. Their dysfunction adds to the symptomatology and pathophysiological process of FGID. FGID-related alterations in intestinal permeability, motility, and visceral hypersensitivity are significantly influenced by ion channels. Channelopathies are caused by genetic mutations and the abnormal functional expression of ion channel subunits. Constipation and diarrhea have been linked to mutations in the ABCC7/CFTR gene. IBS is instead linked to mutations in the SCN5A gene. On the other hand, hypersensitivity and visceral pain in sensory nerves are caused by mutations of the transient receptor potential superfamily’s TRPV1 and TRPA genes. The discovery of a connection between channelopathies and FGIDs opens up new avenues for the identification of novel direct therapeutic targets for specific channelopathies, which has significant implications for the diagnosis and management of FGIDs [84,85].

The heterogeneity observed in FGIDs may be explained by several causes giving rise to distinct disorder manifestations, hence accounting for the observed variety in research outcomes. To better understand the underlying pathogenic mechanisms, more research is necessary.

Table 1 shows evidence that inflammation is involved in the pathogenesis of FGIDs including post-infection bowel syndrome (PI-IBS), a condition that presents after an acute gastroenteritis episode (viral, bacterial, or protozoal) in individuals who do not have previous IBS symptoms, and eosinophilic esophagitis (EoE) [86,87]. The pathophysiological process of PI-IBS involves pathogenic organisms altering the gut microbiota, leading to reduced diversity and an increased Firmicutes/Bacteroides ratio [13]. PI-IBS has shown severe disturbance to the core microbiome (Firmicutes, Bacteroidetes, and Actinobacteria) and a 12-fold rise in Bacteroidetes with a reduction in Firmicutes and Clostridiales when compared to healthy persons. This also causes modifications in bile acid absorption, resulting in diarrhea [77,88,89].

Table 2 and Table 3 show evidence that inflammation is involved in the pathogenesis of FD and IBS.

## 9. Discussion

Many pieces of evidence suggest that FGIDs are very heterogeneous regarding risk factors and disease severity and that, moving forward, the classification and individualized treatment of FGIDs will require a more comprehensive understanding of these disorders than is currently possible. For example, the Rome Foundation Global Epidemiologic Survey’s results demonstrated that patients with FGIDs have a worse quality of life as the number of overlapping FGIDs increases, which may be due to mental disorders and a higher level of activation of neural mechanisms that are present in those patients [111]. Historically, IBS has been classified, together with chronic fatigue and pain syndromes, within “functional somatic disorders” [112]. Somatic symptoms that cannot be attributed to the underlying physical disorders are frequently met in clinical practice, particularly in a primary health care setting. It has been observed that these symptoms are usually linked to some kind of mental discomfort. As the etiologic factors of these symptoms, chronic stress, few methods for coping with stress, and impaired well-being have been suggested [112,113]. The fact that contradicts this theory is that only a subset of these individuals exhibit signs of chronic stress, such as low levels of cortisol in the blood and hair [113]. There are two possible explanations for this finding: (1) that a single biomarker is insufficient to capture the variety of pathophysiology pathways that are present in these patients, which implies the use of several biomarkers for their description; or (2) that the sensitivity of conventional biomarkers to indicate variations in the level of the HPA axis’s arousal or suppression is low, which suggests the need to search for new biomarkers. There is also a possibility that subclinical pathology underlies symptoms of functional somatic disorders. The growing body of evidence, although coming from different research areas, indicates that functional somatic disorders, including FGIDs, should be placed into the spectrum of inflammation-mediated disorders (Figure 3).

This theory is supported by several lines of evidence. The first line of evidence is based on the finding that systemic inflammation has an essential role in generating symptoms of functional somatic disorders such as chronic fatigue syndrome. However, the results of the studies are not consistent, which means that these patients may vary in their level of inflammation [114]. Similarly, serum levels of IL-6, a traditional marker of systemic inflammation, were found to be higher in IBS patients compared to controls, but only in one patient subtype (diarrhea predominant), indicating inter-phenotypic variations in the levels of inflammation [115]. Variations in the level of systemic inflammation in patients with IBS are likely to be acquired, that is, influenced by environmental factors [115,116].

The second line of evidence indicates that mental disorders, even those with a high hereditary influence, are inflammatory in their essence [117]. If mental disorders develop early in life, they often lead to the development of somatic comorbidities in the later course. This particularly refers to cardio-metabolic disorders, which are thought to be a result of the allostatic load on multiple biological systems caused by mental diseases and the chronic stress that accompanies them. Conversely, age-associated somatic disorders that are inflammation-mediated are often accompanied by mental disorders, such as anxiety and depression. Taken together, this means that a shared framework is needed to comprehend the intricate phenomena involving both somatic and mental processes. Innovative research approaches that go beyond conventional study designs and new data evaluation methods will be needed to realize these initiatives [118]. An example might be a recent study, performed in patients with mental disorders, where the focus of the research was on the parallel quantification of two markers of systemic inflammation, IL-6 and C-reactive protein (CRP), and markers of an impaired intestinal epithelial barrier, including LPS and lipopolysaccharide-binding protein (LBP), intestinal fatty acid binding protein (IFABP), and calprotectin, to identify patterns of biomarkers that correspond to phenotypes characterized by certain types and severities of mental symptoms [119].

The third line of evidence refers to studies that, considered together, promote the idea that patients with FGIDs may receive inflammatory signals from two different sources—gut mucosal inflammation and systemic inflammation [68]. To quantify the contribution of these two sources of inflammation would be important because variations in the composition of markers of systemic vs. gut mucosae-associated inflammation may have the potential to create a new, more diverse classification of FGIDs, which, in turn, could lead to etiologically-driven treatment (Figure 3). The idea arises, as shown in Figure 1, from the integrative view of chronic inflammation as being a result of the complex interplay between the neuroendocrine, immune, and metabolic pathways, including the role of the gut microbiota, and being omnipresent in human diseases. Another compelling piece of evidence comes from studies showing that alterations in gut microbiota composition and low-grade inflammation are implicated in the pathophysiology of FGIDs, but there is no consistency between the studies in terms of the number and severity of the involved disorders, which may include any combination of gut microbiota perturbations, loss of barrier integrity, genetic predisposition, or innate immune and specific immune responses. There is a lack of consensus on what the exact role of the gut microbiota is in the pathophysiology of FGIDs, and how changes in its composition relate to these conditions. The principle of “disorders quantification” has already been suggested by the “Immunological homunculus” theory. Although FGIDs are considered disorders of the impaired gut–brain axis, the effect of the upper branch of this axis, that is, mental disorders, on the pathophysiology of FGIDs has rarely been measured. An example may be a study where it was shown that psychological stress in younger individuals with FD, who were still free of overt somatic comorbidities, induced HP infection activation in gastric mucosae, but this was not followed by an increase in expression of the mucosal inflammatory markers IL-6 and IL-8 [120]. Another study implies that severe mood disorders are associated with alterations in circulating levels of both markers of impaired intestinal permeability, and markers of systemic inflammation such as CRP [119].

We demonstrated evidence to support the view that factors like the duration and/or intensity of past stress reactions, gut microbiome dysbiosis, and coexisting somatic comorbidities, all may influence alterations in the gastrointestinal tract’s visceral sensitivity and motility. If put in this context, we can expect, e.g., that in young individuals who do not experience stress for an extended period, only FGIDs which result from activation of the central neuroendocrine stress system, such as altered motility, increased visceral sensitivity, or impaired mucosal blood flow, may contribute to symptoms of FGIDs, without significantly affecting gut microbiota composition and mucosal immunity. Conversely, long-term stress is likely to cause the somatic mechanisms of allostatic load, which usually goes hand-in-hand with gut dysbiosis, increasing the probability of developing epithelial barrier impairment followed by eosinophil/mastocyte gut wall infiltrations (Figure 3). Several lines of evidence support this assumption. For example, the evidence shows that chronic stress has a variety of effects on the gastrointestinal tract, ranging from alterations in gut motility to gut wall structural changes [121]. In addition, histological and tissue immune changes, and changes in serum cytokines and immune cell populations, were shown to vary significantly among studies conducted in patients with FD and IBS [76].

The gastrointestinal tract mucosa is colonized by eosinophils, which often co-localize with mast cells, in both homeostatic and inflammatory conditions [122,123,124]. The physiologic role of these innate immune cells is to maintain intestinal epithelial cell homeostasis against the overgrowth of the luminal microflora. Under stress or disease conditions, the burden of cell infiltrates and the level of their degranulation increases, and this is associated with the release of plenty of inflammatory mediators that can result in tissue damage and fibrosis, instead of homeostatic regulation [101]. Alterations in the levels of markers indicating these events can eventually be found in the circulation. Alternatively, as suggested by the “Immunological homunculus” theory, identifying the pattern of serum naAbs could help create the “image” of how a particular body compartment contributes to the expression of certain FGID phenotypes [38].

In some cases, testing for genetic polymorphisms of genes encoding cytokines, epithelial barrier elements, and serotonin signaling pathways, or for the hidden presence of IBD, can help in understanding deviations from the expected levels of soluble biomarkers (Figure 3) [69,79]. In this respect, studies have shown that the genetic susceptibility for bowel inflammation, as in patients with IBD, is associated with major changes in the gut microbiome composition, and is less likely to depend on variations in environmental and lifestyle factors [125,126,127]. On the contrary, in patients with IBS or those with obesity-related metabolic disorders, modifications in the composition and metabolic activity of the gut microbiome are discrete but subjected to the influence of behavioral factors like diet and sleep quality [128,129]. These results indicate that if we want to achieve a better understanding of the heterogeneity of the clinical expression of FGIDs, data would be necessary to describe a wider patient context. Further confirmation for this statement comes from a study in which it was shown that the gut microbial metabolites may regulate sleep duration through the influence on circadian gene expression, while low sleep quality results in gut dysbiosis and metabolic disturbances due to activation of the HPA-axis [129].

Atopic and allergic conditions, including atopic dermatitis, food allergies, atopic asthma, and allergic rhinitis, are all mediated by a Th2-type immune reaction, which involves the IgE antibody response, IL-4 and IL-5 cytokines, and tissue infiltration with eosinophils and mast cells [130]. It is not surprising that these cells participate in gastrointestinal pathologic conditions, considering the predomination of the humoral IgA antibody response in mucosal immunological reactions. Additionally, there is a link between IgA deficiency and increased IgE antibody production and susceptibility to allergic and autoimmune diseases [131,132]. However, based on the available evidence, it is likely that, in the case of FGID patients, eosinophil and mastocyte gut infiltration is driven by non-Th2-mediated immune responses, which, in conditions of altered gut microbiota, can create an inflammatory environment constituted from Th17 lymphocytes, macrophages, sensitized epithelial cells, mastocytes, and ILC2 innate cells, which drive eosinophil recruitment through the activity of IL-5 [69].

An FGID patient profiling, based on using both laboratory and clinical data, would likely provide more clarity on the phenotypic clustering of these patients. This knowledge could allow for precise treatment, taking into account that FGID patients with, e.g., obesity-related and age-related asthma, which is regarded as T2-low, might have different serum cytokine profiles from young FGID patients with atopic asthma, which is T2-high [88]. Finally, taking data on comorbidities and patient health history could distinguish the origin of increased serum IL-5—either from the gut or distant sources.

Data from current research suggest that the composition and serum concentrations of cytokines and other inflammatory markers are influenced by various factors that characterize patients with gastric symptoms [125,126]. A conclusion that arises is that identifying distinct FGID patient subgroups and placing them within the spectrum of inflammation-mediated chronic disorders would be of the utmost practical importance (Figure 3). This approach will require a comprehensive patient description, using variables such as age, sex, HP infection, the degree of mucosal inflammation, gastrointestinal symptoms, behavioral habits, serum cytokines, certain gut microbial metabolites, and markers of epithelial barrier degradation, and gastrointestinal and extra-gastrointestinal comorbidities. To determine patient subgroups, new methods for data analysis are needed that are based on data integration or clustering, but they are already widely available [133].

As can be seen in Figure 3, at one end of the spectrum the condition may consist of isolated gut disorders, such as those in young adults with unfavorable lifestyles but who still lack other comorbidities. In this case, markers of gut epithelial barrier damage and gut microbial metabolites are expected to be more prevalent in serum than indicators of systemic inflammation, such as Th1-, Th2-, and Th17-type cytokines. These discrete changes in the gut microbiome, and the associated alterations in microbial metabolites, could be responsible for dysfunctional gut motility and visceral hypersensitivity [134]. In conditions with more intensive gut-related inflammation, accompanied by eosinophil and mast cell infiltrations, the composition of serum-soluble factors changes, and these mediators become responsible for the gut nerves and nociceptive receptors’ sensitization.

On the opposite side of the spectrum, there may be older individuals who are obese and have cardio-metabolic disorders. These disorders are associated with both changes in the gut microbiome and increased systemic inflammation, but they still fall within the category of people with low whole-body entropy and just moderate allostatic load. In this case, central neuroendocrine and systemic inflammatory signals, or signals from the outside, have the primary role in sensitizing gut nerves. At the last position, there may be conditions involving severe gut-related and systemic inflammatory responses, such as in individuals with renal failure and CV comorbidities [135].

## 10. Future Perspectives

FGIDs were once thought to be psychosomatic illnesses or motility abnormalities. The term “disorders of the gut–brain interactions” was introduced to reflect the current knowledge of these conditions, which is based on the bio-psycho-social model of chronic diseases. We went a step further in this narrative review, presenting these disorders as a spectrum of phenotypes in which symptoms are attributed to varying amounts of systemic and local gut neuro-inflammatory signals, which are dependent on the social, behavioral, and clinical context of an FGID patient at the time of observation. Drawing on our prior research experience, we suggest that we can gain a new understanding of these disorders with significant therapeutic implications by clustering these patients into distinct phenotypic subgroups and integrating clinical and psychosocial variables with markers of altered gut microbial metabolite production, epithelial barrier dysfunction, and inflammation. This method may lead to a new classification of FGIDs, which will likely inform individualized care and generate creative, original hypotheses that will support additional studies in this field.

## 11. Conclusions

The two most prevalent FGIDs that negatively impact patients’ quality of life and place significant pressure on the healthcare system—particularly family physicians, who treat patients with the symptoms above most frequently—are FD and IBS. The pathophysiology of these disorders is still unclear even though the body of information supporting it is expanding daily. Moreover, the pathophysiology differs between patients. It is established that individuals with these disorders differ from one another, and the inability to precisely identify the mechanism underlying the emergence of symptoms in each of the groups makes therapy even more challenging today. The persistent presence of low-level inflammation has been linked to several cytokines and inflammatory cells, and this is believed to be a fundamental pathophysiological mechanism. The gut–brain axis concept was developed to provide a fundamental framework for the intricate relationships that exist between the neurological system and intestinal processes, including the important role of gut microbes. Systemic chronic inflammation, neuroendocrine disorders, metabolic changes, and gut microbiota dysbiosis are the basis for the emergence of many chronic diseases, the prevalence of which increases with aging and greater exposure to stress. By grouping patients with FGIDs into distinct phenotypic subgroups and combining clinical and psychosocial variables with markers of altered gut microbial metabolite production, epithelial barrier dysfunction, and inflammation, we can obtain a new understanding of these disorders that has crucial therapeutic implications. This approach might result in a new classification of FGIDs, which would probably guide tailored treatment and inspire innovative, creative theories.

## Figures and Tables

**Figure 1 biomedicines-12-00702-f001:**
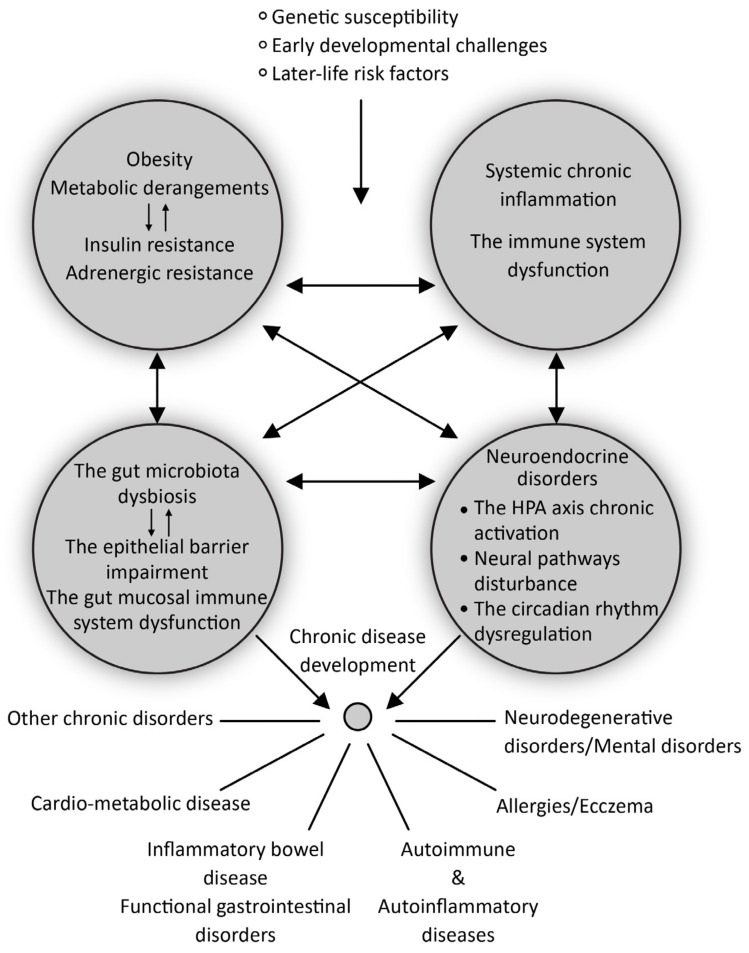
The interplay of systemic chronic inflammation, neuroendocrine disorders, metabolic changes, and gut microbiota dysbiosis in chronic disease development including FGIDs.

**Figure 2 biomedicines-12-00702-f002:**
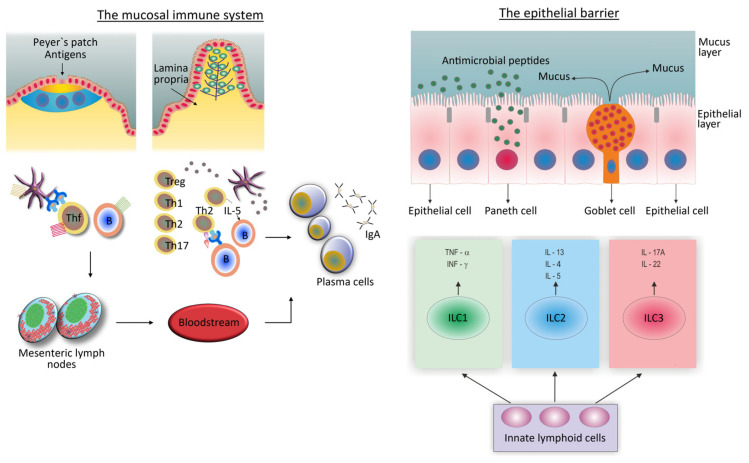
Dual structure (inductive and effector sites) of the gut mucosal immune system.

**Figure 3 biomedicines-12-00702-f003:**
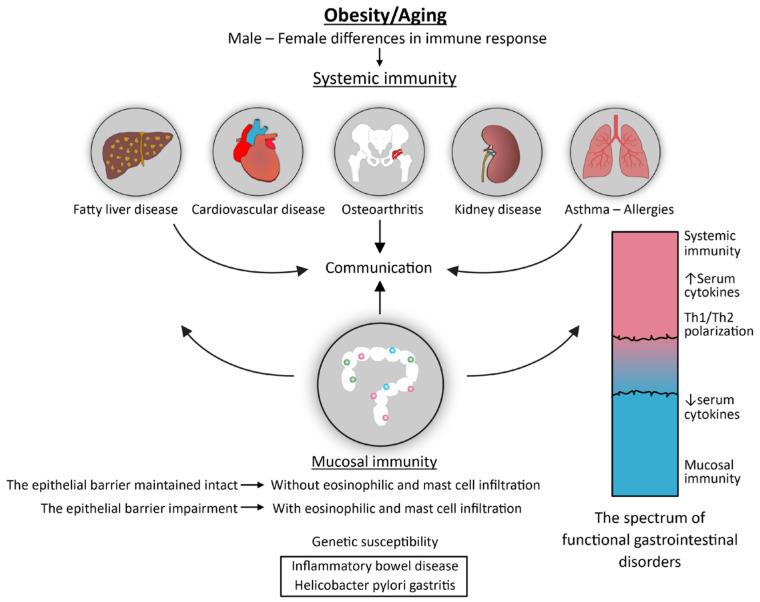
The spectrum of inflammation-mediated chronic disorders communicating through systemic immunity, mucosal immunity, and aging-associated obesity.

**Table 1 biomedicines-12-00702-t001:** Evidence that inflammation is involved in the pathogenesis of FGIDs including PI-IBS and EoE.

Post-Infection Bowel Syndrome
Mast cells, eosinophils, T lymphocytes and macrophages release inflammatory factors (TNF-α, IFN-γ, IL-1, 6, and 8), as well as histamine, leukotriene, and 5-HT, leading to increased intestinal permeability via the reorganization of proteins associated with tight junctions and visceral hypersensitivity. The molecule 5-HT is a crucial neurotransmitter that has a major impact on the functioning of the brain–gut axis, and it is released by ECs. Pain perception brought on by colon distention may be amplified by a long-term high concentration of 5-HT binding to the 5-HT3 receptors on the nociceptive neurons of the vagus in the colorectal mucosa [90,91,92].
Th1 and Th2 cytokine expression varies in the intestinal mucosa, suggesting that the PI-IBS was caused by an immune dysregulation mechanism. Th1-derived cytokine expression (IFN-γ) is increased while Th2-derived cytokine expression (IL-10) is decreased, indicating a shift towards Th1 immunodominance which may lead to chronic low-grade inflammation [93,94].
T helper 17 (Th17) polarizations are observed in IBS, and adenosine and its receptors are involved in inflammation by promoting the Th17 polarization of CD4+ T cells [95].
**Eosinophilic Esophagitis**
EoE is characterized by extensive eosinophilic inflammation causing esophageal-function abnormalities [96].
Epithelial cells and DCs release cytokines (IL-25 and IL-33), and thymic stromal lymphoprotein (TSLP) leading to the activation of invariant natural killer T (iNKT) cells, adaptive CD4+ effector memory Th2 cells, and ILC2. The predominant reaction is the Th2 immune response, secreting IL-4, IL-5, IL-13, IL-15, eotaxin-3, and periostin [97,98].
IL-5 causes eosinophils to multiply and extend from the bone marrow to all layers of the esophagus. They then degranulate, triggering the release of various molecules linked to the remodeling of tissue [99].
Eotaxin-3 is overexpressed causing esophageal mucosal inflammation as a result of environmental antigens.Mast cells can potentially regulate the disease—number of mast cells has been shown to be large even in clinical remission [100].
TGF-β causes mucosal remodeling and smooth muscle dysfunction [101].

**Table 2 biomedicines-12-00702-t002:** Evidence that inflammation is involved in the pathogenesis of FD.

Inflammation in FD
Levels of systemic cytokines and eosinophil infiltration in patients with FD, especially those with PDS, are increased. The eosinophil–mast cell axis secretes chemical mediators influencing visceral hypersensitivity and gastrointestinal motility and leading to neuromuscular and epithelial dysfunction [102,103].
Eosinophil infiltration causes low-grade inflammation in up to 40% of FD patients. When cells degranulate, symptoms occur, along with impaired mucosal integrity and structural and neuronal abnormalities [104].
Unregulated or disrupted activation of mast cells can interfere with gut homeostasis, causing tissue dysfunction, and increasing inflammation [73].
Studies have shown increased levels of TNF-α, IL-1β, and IL-6, all of which are associated with the Th17 pathway. The proliferation and inflammatory activity of Th17 and ILC2 populations may reduce Treg and innate lyphoide type 3 cell (ILC3) populations if homeostasis is disrupted in FGID patients [69].
The observational studies confirm a close association between FD and asthma. This association is especially reasonable when asthma is viewed as an umbrella diagnosis for several diseases with distinct and interrelating inflammatory pathways [87,88].
Contents flow through the mucosa due to enhanced duodenal mucosal permeability. The immune cells causing low-grade mucosal inflammation are the ones that identify them. The enhanced permeability of the duodenal mucosa and weakened epithelial barrier are the results of inflammatory cells’ release of histamine, tryptase, and cytokines, which modify submucosal afferent neurons [75,105,106,107].

**Table 3 biomedicines-12-00702-t003:** Evidence that inflammation is involved in the pathogenesis of IBS.

Inflammation in IBS
Higher serum levels of TNF-α and IL-17 were found to be negatively connected with quality-of-life scores and to be correlated with IBS patients’ discomfort and severity of symptoms [80].
Some studies found no relationship between the severity of the overall symptoms and the expression of the cytokines. IBS patients had higher serum levels of the pro-inflammatory cytokines IL-6 and IL-8 and lower serum levels of the anti-inflammatory IL-10. The correlation between certain clinical symptoms and inflammatory cytokines implies that immunological activation might be significant for diarrhea-predominant IBS patients [108,109,110].
Studies reveal higher numbers and volume of mast cells in IBS patients compared with healthy controls (mast cells counts and density vary among studies and within different segments of the intestine). Mast cells’ mediators modify enteric nerve and motor function, and they have a role in the pathophysiology of IBS. DCs (elevated in some patients with IBS) also contribute to the pathophysiology by inducing visceral hypersensitivity, activating the microcirculation, and prolonging intestinal activity. They can release corticotropin-releasing factor (CRF), which causes changes in visceral hypersensitivity and intestinal motility [80].
Neuroinflammation plays a role in the pathogenesis of IBS through the “gut–brain” axis, altering neuroendocrine pathways and glucocorticoid receptor genes, resulting in a generalized pro-inflammatory phenotype [79].
Pro-inflammatory bacterial species such Enterobacteriaceae are more prevalent than Bifidobacterium and Lactobacillus. The Firmicutes/Bacteroidetes ratio has been changed. These changes can cause a dissolving of mucosal glycoproteins in the intestinal barrier, causing leakage and creating favorable conditions for prolonging the inflammatory state [82].

## Data Availability

Not applicable.

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
