# Peer review of "Putting Functional Gastrointestinal Disorders within the Spectrum of Inflammatory Disorders Can Improve Classification and Diagnostics of These Disorders"

_biomedicines, 2024, doi:10.3390/biomedicines12030702_

Round 1
Reviewer 1 Report
Comments and Suggestions for Authors
1.Overall, it was well organized and the authors wrote down what they wanted to say smartly
2.One of the most important points in FGID is, after all, intestinal microbiological regulation. It would be nice to faithfully explain and supplement the relationship between FGID and inflammation by intestinal microbial control.
3.The importance of ion channels related to FGID regulation is increasing. It would be nice to summarize and explain the ion channels commonly related to FGID and inflammation control.
Reviewer 2 Report
Comments and Suggestions for Authors
This is an interesting article by Šojat Dunja et al.
I would like to address a small number of suggestions to you.
General recommendations
Please check all the text for spelling mistakes.
Abstract
Page 1, line 19
Please explain an abbreviation FGID
Page 1, line 34
Keywords: FGIDs must be added as keywords
Page 1, lie 43
Please rephrase this sentence "There was a revolutionary discovery when, four decades ago, it was realized that colonization.."
Page 2, line 79 The sentence "In this review paper, we intend to promote a hypothesis that FGIDs" has confused me. Is this review article or article hypothesis article?
Page2.
The section 2 "Autoinflammatory and autoimmune disorders" has been unclearly written. In autoinflammation and in “sterile inflammation” neutrophils play an important role. Moreover, after description of NETs in 2004, the role of neutrophils was upgraded.
Please rewrite this section.
The section 3, "The role of inflammation in the pathogenesis of other chronic diseases"
Page 3. line 24 "various stimuli, not just microorganisms" those stimuli are Damage-associated molecular patterns (DAMPs), the endogenous danger molecules that activate the innate immune system through the pattern recognition receptors (PRRs). Please analytically describe PRRs signaling.
Page 3, line 142 It is well documented that IL 17 is also synthetized by other cells type such as neutrophils. Please complete the sentence "Cytokines that are preferentially generated by this lymphocyte subtype are cytokines of the IL-17 family".
The section 5.
To discuss about the interplay between the neuroendocrine, immune, and metabolic pathways it is necessary to write about natural autoantibodies as they are described in immunological homunculus and immunculus theories.
Page 10, line 434
To describe proinflammatory factors such as TNF-α, IFN-γ, IL-1, 6 and 8, authors speculate that mast cells, eosinophils, T lymphocytes, and chromaffin cells release those inflammatory factors. Why don’t you refer to macrophages or neutrophils? Are chromaffin cells able to synthetize TNF-α or IL-1?
The quality of all figures must be improved.
It is necessary to add some tables to the review article.
Round 2
Reviewer 2 Report
Comments and Suggestions for Authors
Dear authors,
Thank you for detailed responses to my all comments.
There are some empty lines in the text.
I suggest deleting lines: line 89 (page 2), line 204 (page 4), line 219 (page 5), line 475 (page 11).
Moreover, please correct word "in-flammation" in the reference 18 and the gap in reference 81.